# Preserving precise choreography of bonds in Z-stereoretentive olefin metathesis by using quinoxaline-2,3-dithiolate ligand

Łukasz Grzesiński[1], Maryana Nadirova [1], Jannick Guschlbauer [1], Artur Brotons-Rufes[2], Albert Poater [2] ✉, Anna Kajetanowicz [1] ✉ & Karol Grela [1] ✉

The Z-alkene geometry is prevalent in various chemical compounds, including numerous building blocks, fine chemicals, and natural products. Unfortunately, established Mo, W, and Ru Z-selective catalysts lose their selectivity at high temperatures required for industrial processes like reactive distillation, which limits their synthetic applications. To address this issue, we develop a catalyst capable of providing Z-alkenes with high selectivity under harsh conditions. Our research reveals a dithiolate ligand that, stabilised by resonance, delivers high selectivity at temperatures up to 150 °C in concentrated mixtures. This distinguishes the dithioquinoxaline complex from existing Z-selective catalysts. Notably, this trait does not compromise the new catalyst's usability under classical conditions, matching the activity of known stereo-retentive catalysts. Density Functional Theory calculations were employed to understand the reaction mechanism and selectivity, and to investigate the poisoning that the catalyst may undergo and how it competes with catalytic activity. Furthermore, the quinoxaline-based catalyst enables the valorisation of bio-sourced alkene feedstocks and the production of agricultural sex pheromones for pest control.

Stereocontrolled synthesis emerged from chemists' need to create organic molecules with precise three-dimensional arrangements. In the context of alkene preparation, it is desirable to control the configuration of the newly formed C–C double bonds, as the E- and Z-alkene isomers often manifest distinct properties. Interestingly, not only simple physicochemical properties such as melting or boiling temperature, but also the biological function of E and Z-isomers can be very different. One such examples are fatty acids, where Z-configured ones are vital components of edible plant oils, while E-fatty acids are believed to contribute to obesity, cardiovascular diseases, and diabetes[1]. Similarly, E/Z-geometric isomers of insect sex pheromones have often different effects on mating behaviour[2,3]. Different olfactory sensation of E and Z-geometric isomers are also well documented, and often one isomer has distinctly more enjoyable (to humans) scent than the other[4]. In this context, it is considered that in many cases Z-isomers have a more pleasant smell and are more natural than E-ones. For instance, (Z)-hept-4-enal isomer possesses a pleasant cream-buttery scent, whereas the E-isomer is known for its pungent putty-like odour[5]. Similarly, Z-configured macrocyclic musk are considered one of the most valued in the flavour and fragrance industry[6]. These compounds can be acquired from animals (e.g. civetone from *Civettictis civetta*, African civet) or synthesised. In the latter case, catalytic olefin metathesis[7–15] can be considered as one of the favourite methods to produce unsaturated musk via ring-closing metathesis macrocyclisation (mRCM)[16]. Unfortunately, preparation of medium and large rings by RCM of unbiased dienes[17] was usually conducted under the

[1]Biological and Chemical Research Centre, Faculty of Chemistry, University of Warsaw, Żwirki i Wigury 101, 02–089 Warsaw, Poland. [2]Institut de Química Computacional i Catàlisi and Departament de Química, Universitat de Girona, c/ Mª Aurèlia Capmany 69, 17003 Girona, Catalonia, Spain. ✉e-mail: albert.poater@udg.edu; a.kajetanowicz@uw.edu.pl; klgre@uw.edu.pl

so-called high-dilution conditions (1–10 mM), making mRCM impractical from the industrial perspective due to high costs of utilising and disposing of very large volumes of organic solvents needed in the synthesis[16]. Recently, using a polymerisation-depolymerisation metathesis reaction under reactive distillation conditions, we pioneered with an efficient mRCM technique producing a number of unsaturated musk as mixtures of *E* and *Z*-isomers at concentration of 200–700 mM[18–20].

## Results and discussion

Because a number of important musk molecules possess *Z*-configured double bonds−profiting from well-established and recently commercially available *Z*-selective molybdenum[21,22] and ruthenium[22,23] catalysts −we decided to test whether the above high-temperature polymerisation-depolymerisation process can be conducted in a stereoselective fashion allowing the synthesis of musk macrocycles as valuable *Z*-isomers (Fig. 1A). As a model transformation in our study we selected mRCM of diene **1** leading to 16-membered lactone **2**−a macrocyclic musk that was previously obtained under classical high-dilution conditions[17,24–27]. Initial tests showed that diene **1** placed under vacuum in a distillation apparatus and heated to 110 °C in the presence of 2.6 mol% of non-*Z*-selective molybdenum catalyst **Mo1** (recently conveniently available as paraffin pellets[28]) in PAO-6 (a type of synthetic paraffin oil used in car engines) used as a diluent, produced a distillate containing (*Z/E*)-**2** (92% of yield) as a 18:82 mixture of *Z/E*-isomers (Fig. 1A). Encouraged by this result, we decided to test the *Z*-selective molybdenum monoaryloxide pyrrolide (MAP) catalyst **Mo2** (also conveniently available as paraffin pellets)[29], in hope of obtaining selectively (*Z*)-**2** using the same reactive distillation method. Using the previously elaborated conditions (PAO-6, vacuum, 110 °C, 8 h), we were pleased to see that lactone **2** was produced in a high isolated yield of 93%. Unfortunately, despite the use of the *Z*-selective catalyst, the *E*-cycloalkene was again formed with a significant preference (*Z/E* = 30:70), which is rather typical to standard, not *Z*-selective catalysts, such as **Mo1** and not to **Mo2**[29].

Similarly, another commercially available *Z*-selective catalyst, **Mo3**, led in high-concentration mRCM distillation to macrocycle **2** in a very satisfactory yield of 87%, albeit again with a rather disappointing *Z*-stereoselection (*Z/E* = 26:74). Given the highly unsatisfactory selectivity of commercial *Z*-selective molybdenum complexes, we opted to test the ruthenium family of *Z*-selective catalysts.

Unfortunately, the commercially available *Z*-selective complex **Ru1**[30,31] failed to produce even traces of macrocycle **2**, which can be attributed to its instability under reactive distillation conditions (Fig. 1A). It is worth to mention that this catalyst was very successful in mRCM under classical conditions in 3 mM solution[30]. Interestingly, the stereoretentive complex **Ru2**[32–34]−initially mis-described as *Z*-selective catalysts[32]−produced a larger amount of the desired *Z*-isomer as compared to previously studied *Z*-selective molybdenum-based catalysts. Namely, the mRCM reaction of **1** conducted in PAO-6 at 110 °C with 10 mol% of **Ru2** led to **2** in 52% of yield as a 58:42 *Z/E* mixture (Fig. 1A). However, it was reported that similar macrocyclic musk has been previously prepared using **Ru2** under standard high-dilution conditions in a solution (3 mM in THF, 40 °C, 1 h) offering much higher *Z*-selectivity level (up to 95:5 *Z/E*)[26,35]. Despite the non-perfect selectivity exhibited by **Ru2** (58:42 *Z/E*) in the high-concentration mRCM reaction we have seen this result as the most promising, and therefore decided to focus on the *Z*-stereoretentive class of Ru olefin metathesis catalysts[32,33,36,37]. Apparently, rather harsh conditions of high-temperature reactive distillation require significantly more stable catalysts to allow high stereocontrol in the metathesis step. The mechanism of stereoretentive catalysts degradation in the presence of terminal olefins via 1,2-sulphide shift was studied in detail[38]. According to Hoveyda et al., the reactive alkylidene-ruthenium species formed due to the Chauvin mechanism[39] are attacked by the sulphide anion of the dichlorocatecholthiolate ligand placed opposite to the NHC ligand (trans effect). This 1,2-sulphide shift leads to the generation of catalytically inactive S-ruthenium complex (Fig. 1C)[38]. Logically, we reasoned that by replacing the strong nucleophilic thiocatecholate dianion **T** (such as in **Ru2**) with a weaker nucleophile stabilised by resonance, like in Wang's paper[40], may perhaps render such designed disulphide ligand less prone to trigger catalysts decomposition via 1,2-sulphide shift mechanism. At the same time, we were afraid that the higher stability can come at the cost of lower catalyst activity and selectivity, as it was reported before[40].

Based on basic textbook knowledge[41] and supported by Density Functional Theory (DFT) calculations, we selected a 2,3-dithioquinaxoline as a precursor for such resonance stabilised ligand (**Q**), leading to a likely thermally more stable complex **Ru3** (Fig. 1B). In fact, nucleophilicity of the sulphide atoms is seen reduced when looking at the corresponding natural charges of each ligand (from −0.44 e⁻ in **T** to −0.39 e⁻ in **Q**). This can be associated with the increased resonance of the new system, already demonstrated in the NBO analysis (see Table S9 in Supplementary information for further details). As a result, the delocalisation of the charge onto the additional aromatic ring of dithioquinoxaline, absolute hardness of the ligand ($\eta$), that is half of the HOMO-LUMO gap[42], increases by 0.035 eV for **Ru3** with respect to **Ru2**, particularly for the further stabilisation of the HOMO orbital (0.078 eV) compared to the one of the LUMO (0.043; Fig. 1D).

## Ligand and catalysts preparation

Based on these considerations, we attempted to prepare the required ligand precursor (Fig. 2A). As a result, a straightforward route has been developed, in which a cheap starting material, benzene-1,2-diamine (**3**) was converted to 1,4-dihydroquinoxaline-2,3-dione (**4**) by an acid-catalysed condensation with oxalic acid (**5**). The obtained dione **4** was further transformed into 2,3-dichloroquinoxaline (**6**) using phosphorus(V) oxychloride. Subsequent interaction of **6** with excess of thiourea led to the formation of non-isolated intermediate (**7**) that was reacted in situ with zinc acetate (Zn(OAc)$_2$×H$_2$O) in the presence of ethylenediamine. The resulting zinc salt **8** was isolated by simple filtration as a canary-yellow solid with excellent yield (90%; see Fig. 2A). It is worth mentioning that such a simple three-step protocol does not require any special conditions and is perfectly suitable to enable large-scale synthesis (tested up to 100 g; for details, see Fig. S1, section 2.2 in supplementary information). Interestingly, by simple hydrolysing of intermediate **7**, it was possible to isolate quinoxaline-2,3-dithiol **9** (more precisely−its tautomeric form−quinoxaline-2,3-dithione, **10**). (Predominance of dithione (=S) form **10** over dithiol (−SH) **9** in a tautomeric equilibrium[43] visibly differentiates this compound from the thiocatechol ligand precursor). The dithioquinoxaline zinc salt **8** was then reacted in THF (RT, 8 h) with the commercially available ruthenium precursor **Ru4** leading to a complex **Ru3** (Fig. 2A). Subsequent filtration through Celite® and concentration under reduced pressure, gave the corresponding ruthenium complex **Ru3** as a brownish solid in 95% yield (note that this step requires strict exclusion of air). Such an obtained catalyst in a solid form can be stored under an argon atmosphere for almost an undefined period of time; however, we noted that in air, ca. 5% of solid **Ru3** is decomposed over 24 h (for details, see Fig. S3, section 2.31 in supplementary information). To evaluate **Ru3** stability in a solution, a quantitative NMR thermal stability study has been performed in THF-$d_8$ (C = 1.0 mM) at 60 and 110 °C under argon. As a result of this measurement, it was found that in a solution **Ru3** exhibits slightly more pronounced thermal stability as compared to **Ru2** (Fig. 2B, for details, see Tables S1, S2, section 2.32 in supplementary information).

Crystals for XRD measurement were obtained by placing a DCM solution of the **Ru3** complex in an NMR tube, then layering *n*-hexane and allowing the solvents to slowly diffuse. Two molecules of the catalyst were found in the asymmetric cell unit. The measurement

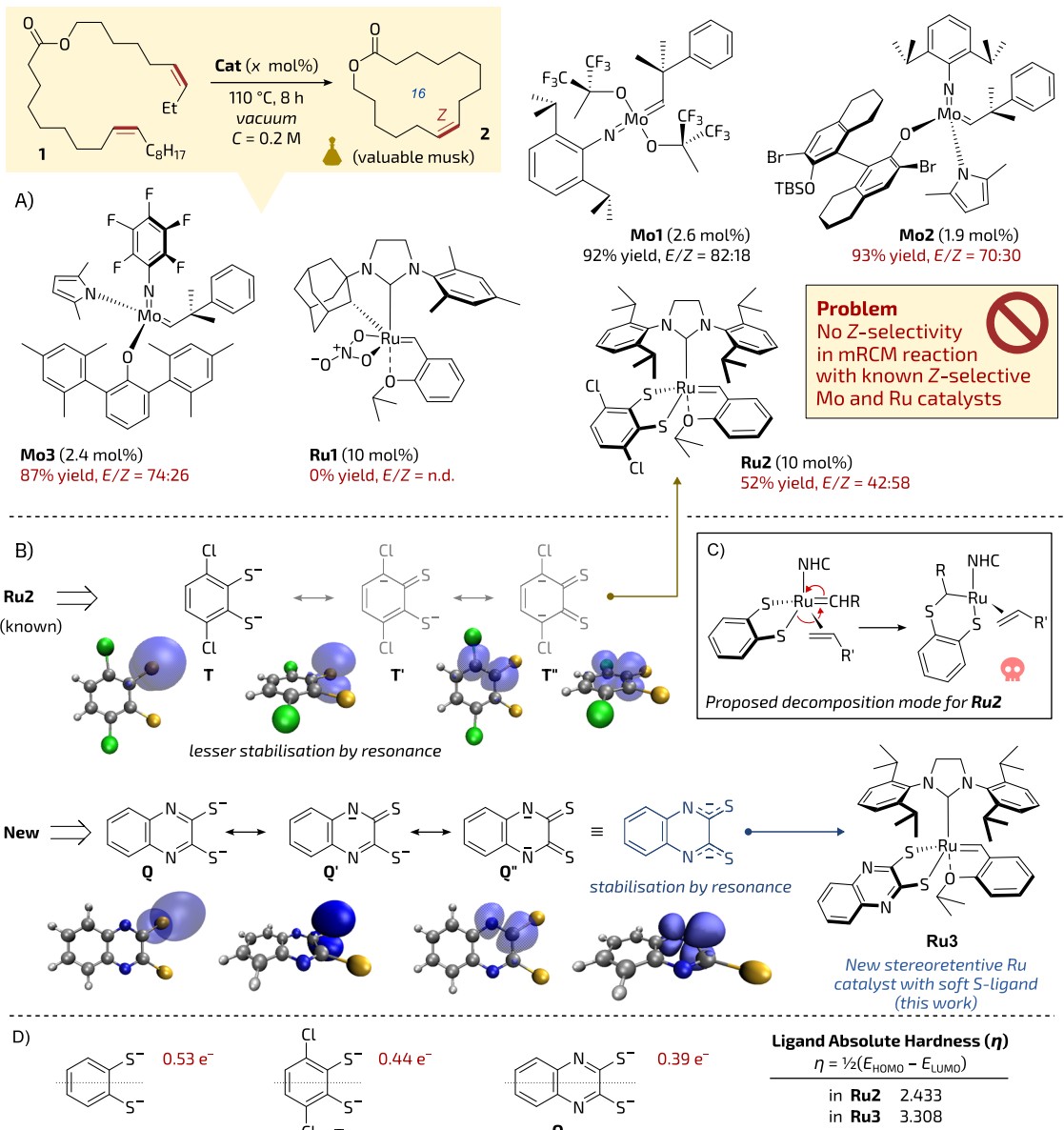

**Fig. 1 | Introduction to the problem. A** Deficiency of selectivity in high-temperature reactive distillation mRCM exhibited by known *Z*-selective catalysts. Yields of isolated pure products. **B** Key resonance structures of known thiocatechol (**T**) and proposed (**Q**) dithiolate anionic ligands. **C** Reported decomposition mode for stereoretentive catalysts via 1,2-sulphide shift (ref. 38). **D** Charges on sulphur (anionic species) in selected ligands and absolute hardness (η) calculated based on frontier molecular orbitals energies (at the M06L/def2-TZVP(PCM(THF))~SDD//BP86-D3/def2-SVP~SDD level of theory) of the ligands in **Ru2** and **Ru3**. n.d. not determined. The skull icon is from Font Awesome', Copyright (c) 2024 Fonticons, Inc. (https://fontawesome.com).

showed similar values of the most important geometric parameters as compared with those from the reported XRD of a thiocatecholate-complex (Table 1)[38]. The distance between the NHC carbon and Ru in **Ru3** was 2.077 and 2.107 Å (two molecules in the asymmetric cell unit). The distances between sulphur atoms and ruthenium were slightly shorter and amounted to 2.3190 and 2.3199 Å for the Ru-S$_1$ bond, and 2.2840 and 2.2818 Å for the Ru-S$_0$ bond. The O-Ru-S$_{0A}$ angle was 169.44° and 168.59°. The key C$_{NHC}$-Ru-S$_1$ angle is amounted to 152.21° and 153.76°.

## Tests in reactive distillation conditions

Having complex **Ru3** in hand, we attempted to test it in the high-concentration *Z*-selective mRCM reaction at high temperatures. To do so, we used the same model diene **1** and conditions as previously (PAO-6, 200 mM, vacuum, 110 °C, 8 h; see Fig. 1A). Surprisingly, **Ru3** performed in the title reactive distillation very well, giving the musk product **2** in 84% isolated yield and—this time—with very good *Z*-selectivity (93:7 *Z/E*, Fig. 3A), which is in striking contrast to **Ru2** providing—under the same conditions—**2** in 52% yield and with much less stereoselection (*Z/E* = 58:42, compare in Fig. 1A). In particular, it was even possible to decrease **Ru3** loading to as low as 0.5 mol% without reducing the yield of the musk product (78%) while maintaining excellent *Z*-selectivity (98:2 *E/Z*, Fig. 3A).

Next, we attempted to obtain a natural product—Yuzu lactone (*Z*)-**11**, a camphor & minty-smelling 13-membered macrocycle isolated from the flesh and peel of the Japanese citrus tree *Citrus junos* Tanaka (Fig. 3A)[44]. Interestingly, this *Z*-configured unsaturated lactone has previously obtained using an indirect diyne metathesis/semihydrogenation sequence[45,46]. More recently, Yuzu lactone was stereoselectively prepared by a direct RCM reaction catalysed by Mo-based MAP complexes; leading to **11** in 46–49% of yield with 69:31 to 73:27 *Z/E* selectivity (reactions were conducted in a high-dilution regime:

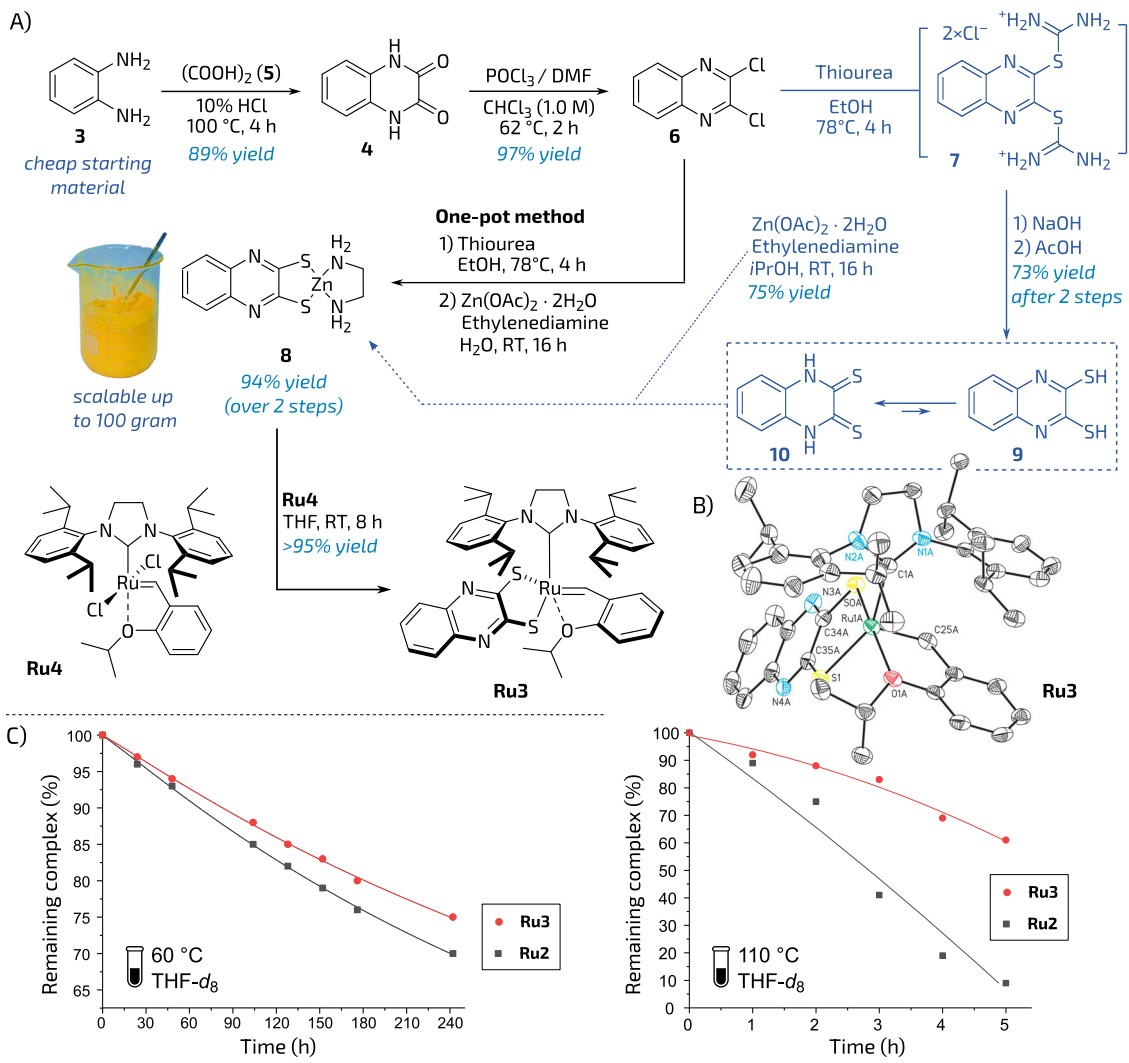

**Fig. 2 | Preparation, solid-state structure, and stability assessment of catalyst Ru3. A** Preparation of ligand precursor (**9**) and catalyst **Ru3**. Yields of isolated pure products. **B** Solid-state crystallographic structure of **Ru3** (25% probability ellipsoids, hydrogen atoms omitted for clarity). **C** Stability comparison of **Ru2** and **Ru3** in THF-$d_8$ solution (C = 0.1 mM) at 60 and 110 °C under argon (anthracene was used as an internal standard). Lines are visual aids only. The test tube icon is from SVG Repo, Items And Tools Icooon Mono Vectors Collection, Public Domain License (https://www.svgrepo.com).

5 mM)[47]. (For comparison, non-*Z*-selective catalyst, **Mo1**, was reported to give in a solution product **11** in 30% yield as 17:83 *Z/E* mixture.)[47] Ruthenium *Z*-selective complex **Ru1** gave in this reaction (7.5% mol%, 3 mM in DCE, 24 h) 40% of yield and *Z/E* = 86:14[30]. On the other hand, stereoretentive Ru complex, **Ru2** (used in 6 mol% loading) in a diluted DCM solution (3 mM) gave **11** in 68% yield and high selectivity (>95% *Z*)[26]. In this context we were pleased to see that only 1 mol% of **Ru3** under the reactive distillation conditions (110 °C, PAO-6, 200 mM) led to Yuzu lactone **11** in isolated yield of 66% and high *Z*-selectivity of 99:1 (Fig. 3A).

To further explore the scope of our stereoretentive catalyst, another experiment was conducted in the presence of 1 mol% of **Ru3**, leading to the isolation of 19-membered macrocyclic ether **12** with satisfactory yield (68%) and similarly high *Z*-selectivity (≥98:2 *Z/E*, Fig. 3A).

Another example of **Ru3** being favoured for the discussed transformation is mRCM leading to Civetone (**13**)–a naturally occurring animal musk isolated from the glands located near the rectum of the African civet (*Civettictis civetta*) and the large Indian civet (*Viverra zibetha*). We were pleased to see that with a loading as low as 1.0–0.5 mol% our catalyst gave macrocyclic ketone (*Z*)-**13** in 84–76% isolated yield with perfect stereoselectivity. For comparison, the

benchmark system, **Ru2**, under the same conditions (110 °C, 0.5 mol%) gave only 13% of **13** and worse selectivity (83:17 *E/Z*, Fig. 3A). Importantly, **Ru3** remained impressively selective (96:4 *Z/E*, see insert in Fig. 3B) in mRCM production of Civetone even at an increased temperature (up to 150 °C). Finally, high-oxidation-state Mo-based alkylidene catalyst **Mo1** under reactive distillation conditions failed to produce ketone **13** at all (Fig. 3B), probably because of a competing olefination reaction[48,49]. For comparison, under classical high-dilution conditions, **13** was obtained with 7.5 mol% of *Z*-selective adamantyl-complex **Ru1** in 50% yield as a 68:32 *Z/E* mixture[30] (with optimised *Z*-selective catalyst of the same family Civetone **13** was obtained in improved selectivity (95% of *Z*) but in reduced yield (36%))[50]. It should be noted that (*Z*)-Civetone (**13**) has also been obtained in 44% yield and in perfect *E/Z* selectivity of 95:5 in a continuous flow reactor using a variant of **Ru1** (7.5 mol% of catalyst; 3 mM in DCE, 3 h, 70 °C)[31].

## Tests under classical conditions

Being surprised of such high *Z*-selectivity levels exhibited by **Ru3** that were preserved even at as high temperature as 110–150 °C during reactive distillation, we attempted a separate set of tests to verify applicability of **Ru3** in reactions other than high-temperature mRCM. First, three model cross-metathesis (CM) reactions were attempted

(Fig. 4A). These transformations were conducted in neat, while one of the products (3-hexene, **15**) was removed under reduced pressure (Fig. 4A). Under these conditions the internal alkenes: (Z)-3-hexen-1-ol, (Z)-6-nonen-1-ol, and (Z)-6-nonenal (**14, 17** and **21**) were dimerised to yield valuable Z-configured functionalised products **16, 18** and **22** highly selectively (96:4 to 99:1 Z/E). Interestingly, one of the products—the bis(aldehyde) **22**—was found to possess a very intense musky fruity scent. These reactions were easy to scale-up (up to 5 g) and required only a small amount of the catalyst (0.5 to 0.01 mol%) to proceed. It was also possible to conduct two different CM reactions sequentially *one-pot*, where the same portion of **Ru3** (0.5 mol%) first dimerised substrate **14** then, in the presence of (Z)-1,4-diacetoxy-2-butene (**19**) promoted a CM reaction leading to bifunctional product **20** in 52% yield in two steps and with high selectivity (Z/E = 98:2, Fig. 4A).

### Table 1 | Selected lengths (Å) and angles (°) of complex Ru3

| Bonds | Length (Å) or angle (°) [a] |
|---|---|
| Ru(1)-C(1) | 2.077(6) |
|  | 2.107(5) |
| Ru(1)-O(1) | 2.321(4) |
|  | 3.318(4) |
| Ru(1)-C(25) | 1.820(7) |
|  | 1.829(5) |
| Ru(1)-S(0) | 2.2840(14) |
|  | 2.2818(13) |
| Ru(1)-S(1) | 2.3190(16) |
|  | 2.3199(13) |
| C(1)-Ru(1)-S(1) | 152.21(18) |
|  | 153.76(16) |
| O(1)-Ru(1)-S(0) | 169.44(11) |
|  | 168.59(11) |

[a]Two molecules in the asymmetric cell unit.

Next, we opted to test the activity and selectivity of **Ru3** under classical conditions (in a solution), similar to those used frequently in stereoretentive catalysis[34]. In this context, the CM reaction of allyl-benzene and (Z)-1,4-diacetoxy-2-butene (**19**) is commonly considered the model reaction. Although THF is the most frequently used solvent with benchmark **Ru2**, we decided to perform a small solvent screening study to check if the same solvent is also optimal for **Ru3**. To do so, EtOAc, toluene, perfluorotoluene, dimethyl carbonate (DMC), 4-methyltetrahydropyran (4-MeTHP), 2-methyltetrahydrofuran (2-MeTHF) and dichloroethane (DCE) were tested at a range of temperatures (40, 60 and 90 °C, for details, see Table S5, section 2.6.1 in supplementary information). The outcome of these studies has indicated that the choice of solvent perceptibly impacts the results of the model reaction: amongst the screened variety, THF has been found to give the best Z/E ratios and yields, rendering DMC the second choice. Thus, using **Ru3** in THF at 40–60 °C substrates such as allylbenzene, 4-methoxy(allylbenzene) and (Z)-6-nonen-1-ol were reacted with **19** yielding the expected products **23, 24** and **25** in yields and selectivity fully comparable to those offered by **Ru2** (Fig. 4B). In the second set of reactions, selected terminal and internal alkenes (allylbenzene, 1-dodecene and (Z)-6-nonenal) were reacted with other common cross-partners, such as (Z)-1,4-butendiol and (Z)-1,4-dibenzyloxy-2-butene in THF leading to products **26, 27** and **28**. Also, in this case, activity and selectivity levels offered by **Ru3** fully matched the one provided by benchmark **Ru2** (Fig. 4B).

Encouraged by results observed for **Ru3** under classical conditions in solution, we moved towards more sophisticated and challenging substrates. Hence, selected API (Active Pharmaceutical Ingredient) derivatives bearing various heterocyclic and Lewis basic groups (known potential chelators to Ru) were tested with **Ru2** and **Ru3** (Fig. 4C). It is worth highlighting that despite high structural complexity represented by the tested pharmaceutical models, starting from estrone derivative **29**, trough β-lactam **30**, an analogue of psychoactive cannabinoid agonist UR-144[51] (**31**), a Sildenafil (Viagra™)[52] derivative (**32**), the catalyst **Ru3** was able to provide the expected products with always exceptional Z-selectivity (98-99%), fully matching

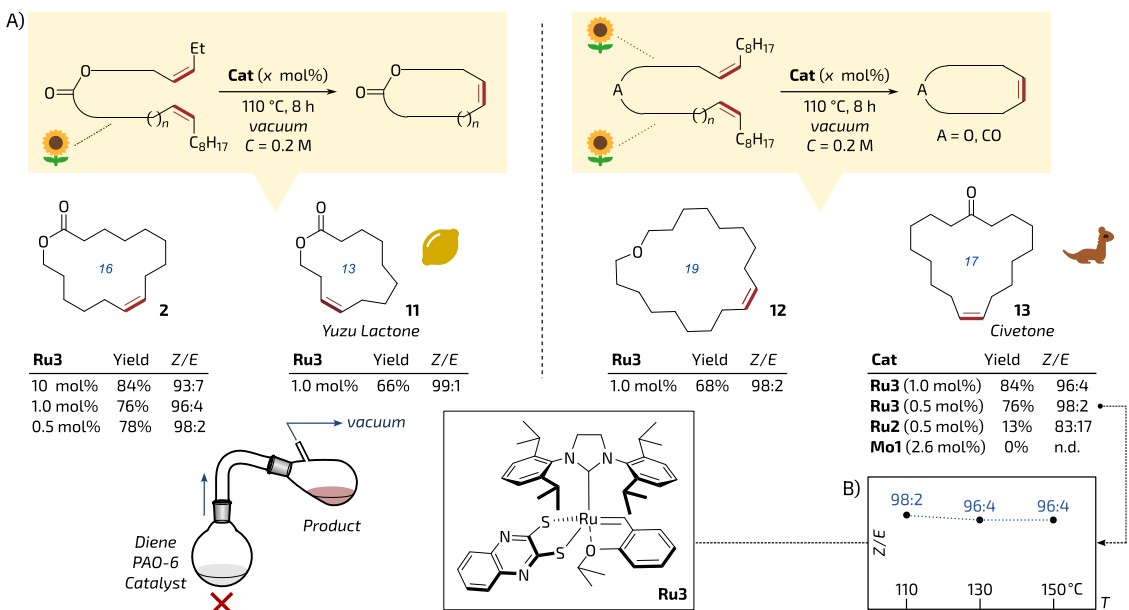

**Fig. 3 | Z-Stereoreoselective reactive vacuum distillation at high concentration. A** Examples of bio-based (partially or fully derived from oleic acid, which is symbolised by a sunflower icon) macrocycles obtained in reactive vacuum distillation at high concentration. Yields of isolated pure products. **B** (in inset) **Ru3** Z-selectivity proven up to 150 °C. The citrus icon is from Font Awesome', Copyright (c) 2024 Fonticons, Inc. (https://fontawesome.com). The sunflower icon is from SVG Repo, Fxemoji Emojis Collection, Apache License and the civeta icon is from SVG Repo, Animals 30 Collection, CC0 License (https://www.svgrepo.com). The distillation set is adapted from A. Sytniczuk, M. Milewski, M. Dąbrowski, K. Grela and A. Kajetanowicz, *Green Chem.*, **2023**, 25, 2299–2304, DOI: 10.1039/D2GC02988J.

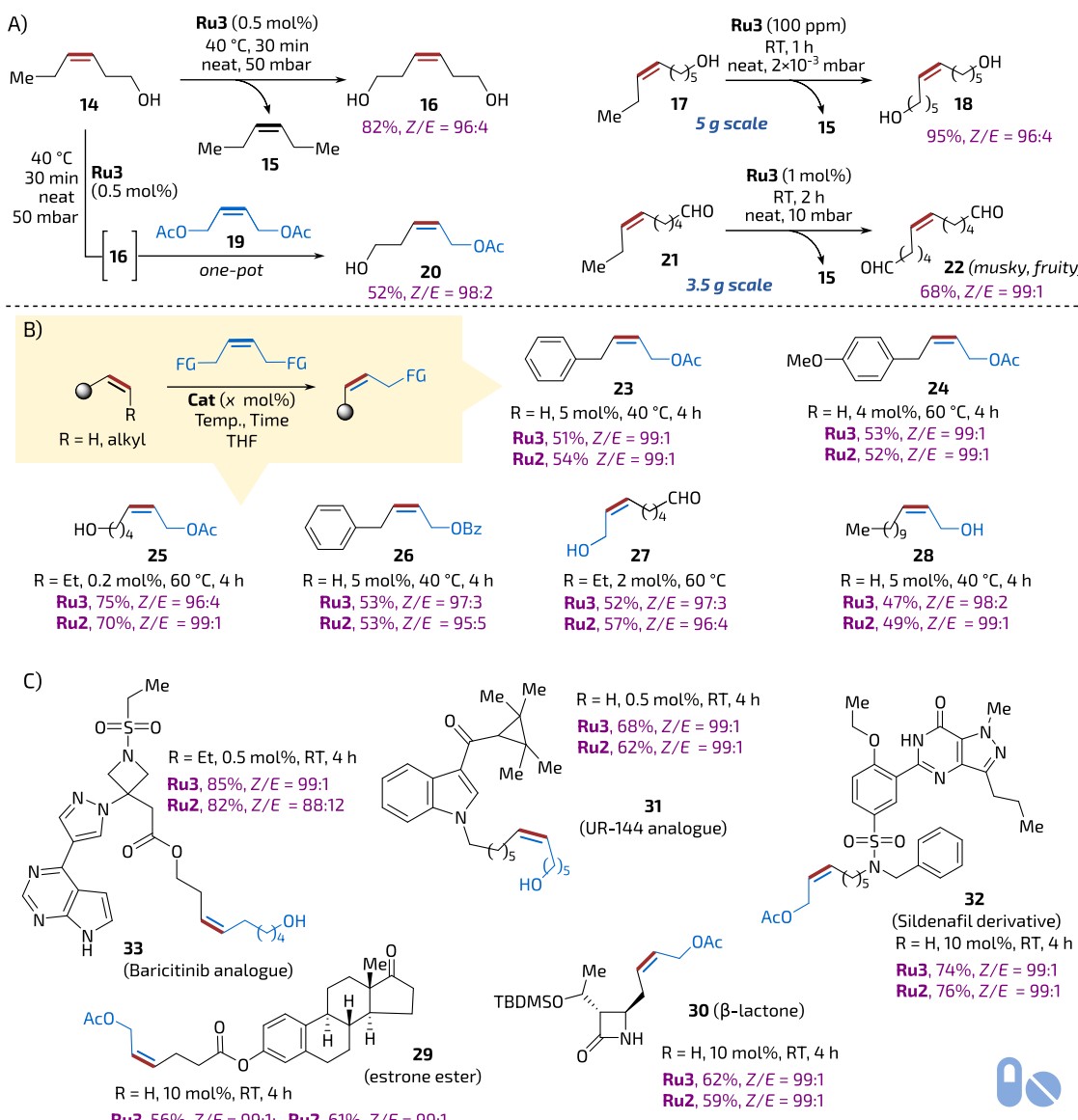

**Fig. 4 | Z-Stereoselective Cross-Metathesis (CM) reactions. A** Z-selective CM made with **Ru3** under reduced pressure without solvent (neat conditions). **B** Comparison of **Ru2** and **Ru3** in Z-selective CM under classical conditions (in THF solution). **C** Examples of complex polyfunctional API-derived molecules obtained in Z-selective CM. Yields of isolated pure products. API active pharmaceutical ingredients. The tablet icon is from Font Awesome', Copyright (c) 2024 Fonticons, Inc. (https://fontawesome.com).

the established stereoselective catalyst **Ru2**. Finally, in the case of complex polyfunctional molecule of Baricitinib[53] analogue (**33**) quinoxaline-derived **Ru2** seems to slightly outdistance **Ru2** in terms of yield and selectivity (Fig. 4C).

Chemical conversion of plant oils by modern catalytic methods is believed vital for developing eco-friendly routes to various building blocks and fine chemicals[54,55]. Therefore, we opted to test the quinoxaline-derived catalyst in Z-stereoretentive self-metathesis of oleic acid methyl ester (Z)-**34**, confronting it with the results obtained previously by Grubbs[56] and Mauduit[57]. It was found that **Ru3** used in 0.1 mol% loading led to 50% conversion of methyl oleate (equilibrium) in less than one hour, thus being more effective than **Ru2** and **Ru5** (however, the specialised, sterically activated catalyst **Ru6**[58] was reported to achieve the equilibrium in even a shorter time and at lower loading: 15 min at 0.05 mol% or 16 h at 0.01 mol%[57]). Importantly, products of this reactions—alkenes **35** and **36** obtained with high stereoselection (Z/E = 99:1) are known as valuable building blocks, being used inter alia in the synthesis of (Z)-Civetone (from diester **36**)[59,60],

and various pheromones (from both **35** and **36**)[31,61]. Catalyst **Ru3** was tested in self-metathesis of oleic acid **37**, and also, in this case, achieved an equilibrium with perfect Z-selectivity at only 0.25 mol% of catalyst loading within 1 h (Fig. 5A).

Pheromones offer a highly promising, biosphere-friendly approach to protect various crops. Unlike chemical pesticides, pheromones target only specific pest species, leaving other insects, especially pollinators (bees and wasps), unharmed[2,3]. Additionally, pheromones are biodegradable and used in small amounts, and therefore do not pose a threat to human health, making them ideal for modern, environmentally conscious farming[62]. However, their synthesis shall be possibly straightforward and fully stereoselective (as the cost of pheromone production shall be kept down to make them applied also in less developed countries. Presently, treating a one-hectare field with a regular pesticide costs between €50 and €150. The bio-control of the same surface using the pheromone costs around €250 according to reference[62]), as the biological action of pheromones is often related to their E/Z-geometrical constitution[2,3]. Therefore, as a

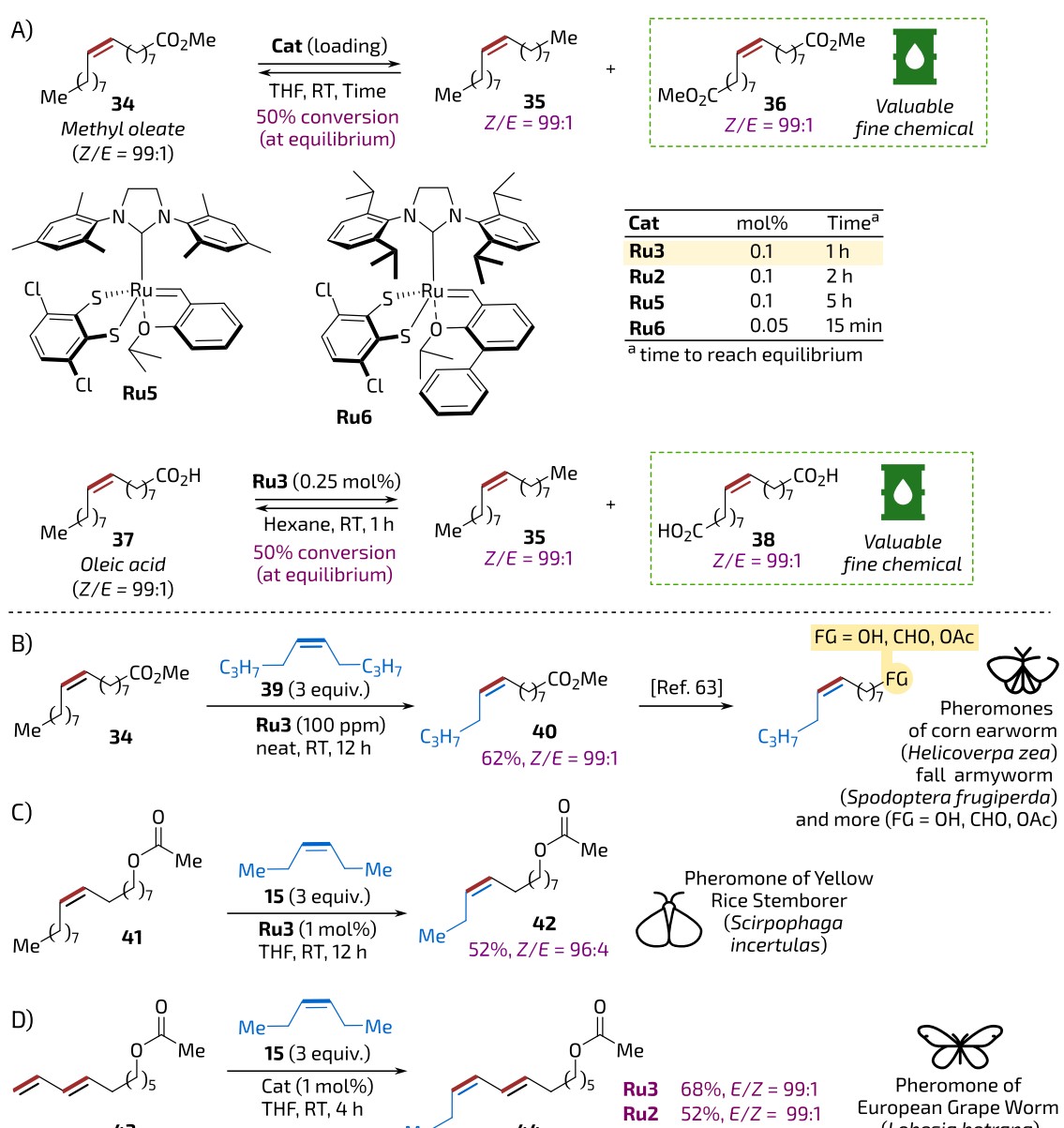

**Fig. 5 | Conversion of fatty acid derivatives into fine chemicals. A** *Z*-selective self-CM of oleic acid and its methyl ester. **B–D** Examples of pheromones or pheromone precursors obtained in *Z*-selective CM with **Ru3** (for conversion of precursor **40** into pheromones, see El-Rabbat and Mangold, ref. 68). The barrel icon is from Font Awesome', Copyright (c) 2024 Fonticons, Inc. (https://fontawesome.com). The moth icons are from SVG Repo, Bugs Insects 2, Nature 3, and Spring Icons Collections, CC0 License (https://www.svgrepo.com).

final test for the usefulness of our dithioquinoxaline-based catalyst, we opted to prepare some well-known pheromones using *Z*-selective metathesis with **Ru3**.

First, we approached the synthesis of a precursor of sex pheromones of fall armyworm (*Spodoptera frugiperda*)[63] oriental tobacco budworm moth (*Helicoverpa assulta*)[64], corn earworm (*Helicoverpa zea*)[65] and similar insects from the *Noctuidae* family that consume a wide variety of crops such as corn, tomato, pepper, rice, peanuts, and more. In a published method, the CM reaction of methyl oleate (**34**) was conducted with 1-hexene and the second-generation Hoveyda-Grubbs catalyst supported on silica. This transformation produced a rather complex mixture of products, including 1-decene, 5-tetra-decene, methyl 9-decenoate and methyl 9-tetradecenoate (**40**)[66]. Obviously, due to the application of non-*Z*-selective catalyst, this method suffered from very low *E*/*Z*-selectivity. Therefore, Pederson and Grubbs used an alternative approach that was based on the reaction of a large excess of 1-hexene (in fact, used as a co-solvent, as a 1:1

mixture with THF) with oleyl alcohol. This process yielded a pheromone precursor−(*Z*)-9-tetradecenol in a 77% of yield, albeit exhibiting non-perfect *Z*-selectivity (*Z*/*E* = 86:14) despite the use of the *Z*-selective catalyst **Ru1** (1 mol %)[67]. In turn, we decided to reinvestigate the methyl oleate route, as the precursor (*Z*)-**40** can be later converted to a number of *lepidopteran* sex pheromones (Fig. 5B). As a result, we were pleased to see that **34** reacted with internal olefin **39** (3 equivalents) to lead (*Z*)-**40** with high selectivity (*Z*/*E* = 99:1) in 62% yield, in presence of only 100 ppm of **Ru3**. It shall be noted that this reaction can be conveniently conducted without a solvent (in neat). Such formed (*Z*)-**40** can be converted to valuable pheromones, such as (*Z*)-9-tetradecenal, (*Z*)-9-tetradecenol and (*Z*)-9-tetradecenol acetate by reduction and acetylation using reported methods (Fig. 5B)[68].

Next, we attempted to do CM between oleyl acetate **41** and internal olefin **15** to obtain a pheromone of yellow rice stem borer (*Scirpophaga Incertulas*), a moth whose larvae causes severe damage to rice throughout the world[69]. The *Z*-selective CM reaction went in this

case easily as well, leading to the formation of **42**[67] as almost pure *Z*-isomer (*Z*/*E* = 96:4, Fig. 5C). For comparison, Pederson and Grubbs, to obtain the same pheromone, used CM between oleyl alcohol and 1-butene. The optimal conditions involved slow bubbling of 1-butene into the reaction solution in the presence of 2 mol % of **Ru1**. Reaction with oleyl alcohol (followed by acetylation of the CM product) led, however, to the formation of pheromone **42** in modest yield (40%) and slightly reduced *Z*-selectivity (77% *Z*)[67].

Finally, we decided to try CM of conjugated diene **43** with **15** to obtain sex pheromone of the European grapevine moth (*Lobesia botrana*)−an environment-friendly agrochemical already used to protect vineyards in Europe[62]. To do so, in a reaction catalysed by 1 mol% of **Ru3** product **44** was formed as a single (7*E*,9*Z*)-isomer with high selectivity (Fig. 5D). Importantly, the industrially used method[70], as well as the improved preparation[62] let to this pheromone as (7*E*,9*Z*) and (7*E*,9*E*) 75:25 mixture of isomers.

**Fig. 6 | Density functional theory (DFT) studies. A** Subsequent cross-metathesis cycles were analysed. **B** Reaction pathway with the corresponding relative Gibbs energies (in kcal/mol) for **Ru2** (in black) and **Ru3** (in green). The transition states correspond to TS1 for the metallacycle formation and TS2 for the cycloreversion steps. In the initial cycle during precatalyst activation, the corresponding cases are labelled with an h- prefix, such as hTS1 and hTS2, with the chelate aperture indicated by hTS0.

**Table 2 | Thermodynamics and kinetics of the 1,2-sulphide shift for different propagating species**

| | from B | | from B' | | from E | | from F | | from G | |
|---|---|---|---|---|---|---|---|---|---|---|
| | Ru2 | Ru3 | Ru2 | Ru3 | Ru2 | Ru3 | Ru2 | Ru3 | Ru2 | Ru3 |
| R | 15.0 | 15.3 | 18.8 | 19.2 | 15.5 | 13.3 | 11.9 | 10.2 | 14.1 | 14.2 |
| TS$_{1,2\text{-shift}}$ | 29.7 | 29.9 | 31.7 | 31.2 | 26.4 | 23.9 | 14.4 | 18.9 | 26.5 | 27.0 |
| P | 20.1 | 20.6 | 17.3 | 20.3 | 12.3 | 11.9 | 1.9 | 2.7 | 19.7 | 20.1 |

All values are given in kcal/mol.
R reactant(s), TS$_{1,2\text{-shift}}$ transition state for a 1,2-sulphide shift, P products.

## DFT study

To analyse possible differences in deactivation of **Ru3** and **Ru2** catalysts via 1,2-sulphide shift we decided to model the CM reaction of allylbenzene and (Z)-1,4-diacetoxy-2-butene which is a popular test reaction in Z-selective metathesis (compare: Fig. 4B). When analysing the deactivation reaction paths, it is crucial to account for the various olefin combinations present. Speciation processes must be considered to accurately assess the deactivation step, as it can occur within any of the CM cycles. Figure 6A illustrates some of these cycles, and while not explicitly investigated, certain non-productive combinations were also taken into account. During the reaction precatalyst **Ru2** or **Ru3**—after dissociation of Ru−O chelate that leads to formation of 14-e⁻ species **A**—undergoes a series of 2 + 2 cycloaddition and cycloreversion reactions[39] forming key propagating intermediates **RuI, RuII** and **RuIII** (Fig. 6A). During each of the consecutive catalytic cycles these intermediates can keep propagating or undergo decomposition via 1,2-sulphide shift (Fig. 1C)[38]. After studying the reaction sequence in silico, we noted no significant differences between **Ru2** and **Ru3** catalysts thermodynamics (Fig. 6B). Although the energy barriers for **Ru3** appear to be slightly smaller, this reduction falls beyond the precision of DFT calculations.

Table 2 presents the thermodynamics of the 1,2-sulphide shift transition state (TS) with different alkene substituent sets during the activation and first cross-metathesis cycle. Similar to the cross-metathesis reaction discussed earlier, the disparities between both systems are, in general, small. However, two notable distinctions emerge in the following scenarios: when an ethylene molecule coordinates with a 14-e⁻ methylidene complex (**E**), the 1,2-sulphide shift occurs more rapidly with the benchmark **Ru2** compared to the **Ru3** system. Another case where these two systems exhibit different behaviour involves the coordination intermediate comprising of methylidene and allylbenzene (**F**). In this instance, the kinetics favour the deactivation of Ru2 by ~5 kcal/mol.

The Z-alkene geometry is ubiquitous in a wide range of building blocks, fine chemicals, and natural products. Because well-established Mo, W, and Ru Z-selective catalysts were found to lose their selectivity at higher temperatures required for industrially interesting reactive distillation mRCM, we opted for the development of a catalyst capable of providing Z-alkenes in high selectivity, also under more harsh conditions. It was found that the new type of stabilised by resonance dithiolate ligand can provide high selectivity at temperatures up to 150 °C in highly concentrated reaction mixtures. Computational studies were employed to assess the observed enhancements conferred by these novel ligands, operating under the assumption that the primary limitation to achieving Z selectivity lies in the propensity for 1,2-sulphide displacement. While no discernible thermodynamic enhancements were identified along the Z-product route, there is evidence suggesting a reduction in the kinetics of the deactivation pathway with our ligand, particularly when considering proper speciation. Furthermore, the concurrent characterisation via parallel DFT supports the notion of a nucleophilic decrease in the sulphur nuclei within the 2,3-dithioquinoxaline ligand structure.

The above attributes distinguish the **Ru3** complex from other Z-selective olefin metathesis catalysts based on Mo, W and Ru. Importantly, this unique trait did not come at the cost of limited general usability under 'classical' conditions. On the contrary, the new complex was found to match the activity of known stereoretentive catalysts such as **Ru2** even in the case of complex polyfunctional substrates. In addition, the catalyst **Ru3** can be used in the valorisation of bio-sourced alkene feedstock such as oleic acid and in the production of sex pheromones used in agriculture for pest control.

## Methods
### Synthesis of Ru3
In a glovebox, to a flask charged with 2,3-dithioqunoxalineethylenodiamine zinc complex **8** (593 mg, 1.86 mmol, 2.0 equiv.)

and **Ru4** (660 mg, 0.93 mmol, 1.0 equiv.), dry THF (50 mL) was added. The reaction mixture was stirred at room temperature for 8 h, concentrated in vacuo, and the residue was diluted in dry DCM (50 mL). The formed suspension was filtered through a Celite pad (0.5 cm high, 3 cm diameter), and the filtrate was evaporated to dryness and co-evaporated with pentane (3 × 10 mL) three times. The product was obtained as a dark brown solid with 0.5 equiv. of DCM per formula unit as co-solvent (768 mg, 0.88 mmol, 95% yield). $^1$H NMR (400 MHz, CDCl$_3$) δ 15.02 (s, 1H), 7.85 (dd, $J$ = 8.2, 1.5 Hz, 1H), 7.79 (dd, $J$ = 8.1, 1.4 Hz, 1H), 7.47 − 7.30 (m, 7H), 7.28 (dd, $J$ = 7.7, 1.6 Hz, 1H), 7.21 (dd, $J$ = 7.7, 1.5 Hz, 1H), 6.90 (d, $J$ = 8.4 Hz, 1H), 6.82 (td, $J$ = 7.4, 0.8 Hz, 1H), 6.73 (dd, $J$ = 7.6, 1.6 Hz, 1H), 6.58 (dd, $J$ = 7.6, 1.7 Hz, 1H), 4.94 (hept, $J$ = 6.2 Hz, 1H), 4.40 − 4.20 (m, 1H), 4.20 − 4.05 (m, 2H), 4.05 − 3.90 (m, 2H), 3.90 − 3.77 (m, 1H), 3.09 (hept, $J$ = 6.8 Hz, 1H), 2.39 (hept, $J$ = 6.7 Hz, 1H), 1.91 (d, $J$ = 6.5 Hz, 3H), 1.39 (d, $J$ = 5.9 Hz, 3H), 1.33 (d, $J$ = 6.8 Hz, 3H), 1.28 (d, $J$ = 7.0 Hz, 3H), 1.26 (d, $J$ = 6.9 Hz, 2H), 1.04 (d, $J$ = 6.8 Hz, 3H), 0.99 (d, $J$ = 6.3 Hz, 3H), 0.92 (d, $J$ = 6.7 Hz, 3H), 0.67 (d, $J$ = 6.6 Hz, 3H), 0.01 (d, $J$ = 6.7 Hz, 3H). $^{13}$C NMR (101 MHz, THF-$d_8$) δ 262.2, 220.9, 171.5, 163.4, 156.5, 150.2, 149.3, 147.6, 1456.0, 142.1, 139.2, 139.0, 138.9, 136.6, 131.1, 130.7, 129.3, 129.3, 128.8, 128.0, 127.4, 126.2, 126.2, 126.0, 125.9, 125.6, 125.3, 125.1, 124.5, 123.0, 115.5, 29.9, 29.4, 29.28, 29.0, 27.3, 26.9, 26.6, 24.2, 22.8, 22.2, 20.9, 20.3, 14.3. HRMS (ESI) Calcd. for C$_{45}$H$_{55}$N$_4$ORuS$_2$ [M + H]$^+$: 833.28553. Found: 833.28593. EA Calcd. for C$_{45}$H$_{54}$N$_4$ORuS$_2$ × 0.5 CH$_2$Cl$_2$: C, 62.49; H, 6.34; N, 6.41; S, 7.33. Found: C, 62.45; H, 6.50; N, 6.27; S, 7.38. IR (cm$^{-1}$) 3061, 2961, 2924, 2866, 2656, 2451, 2323, 2289, 1682, 1589, 1474, 1453, 1440, 1408, 1385, 1364, 1324, 1255, 1236, 2272, 1119, 1094, 1046, 1016, 919, 802, 752, 621, 599, 566, 457, 427.

## Data availability

The experimental data in this study are provided in the Source Data file. The data generated in this study have been deposited in the Zenodo repository under the accession code https://doi.org/10.5281/zenodo.13384825. All additional data were available from the corresponding author upon request. Crystallographic data for the structures reported in this Article have been deposited at the Cambridge Crystallographic Data Centre, under deposition number CCDC 2350437 (Ru3). Copies of the data can be obtained free of charge via https://www.ccdc.cam.ac.uk/structures/. Source data are provided with this paper.

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

## Acknowledgements

The authors are grateful to the OPUS project financed by the National Science Centre, Poland, on the basis of a decision UMO-2019/33/B/ST4/00874. A.P. is a Serra Húnter Fellow, and ICREA Academia Prize 2019, and thanks the Spanish Ministerio de Ciencia, Innovación y Universidades (MICIU) for project PID2021-127423NB-I00 and the Generalitat de Catalunya for project 2021SGR623. The project was conceived and designed by K.G. and Ł.G. Ł.G. synthesised complex **Ru3**, Ł.G., M.N. and J.G. performed the catalytic tests, A.B.-R. and A.P. performed DFT calculations, Ł.G., A.K., K.G. and A.P. contributed to writing the paper and critically commented on the manuscript.

## Competing interests

The authors declare no competing interests.
