## [Peer Review File · Nature Communications]

Preserving Precise Choreography of Bonds in
Z-Stereoretentive Olefin Metathesis by Using
Quinoxaline-2,3-dithiolate LigandREVIEWER COMMENTS

Reviewer #1 (Remarks to the Author):

This work by Grela describes the synthesis of a new, robust and highly Z selective olefin metathesis catalyst. The reported catalyst is the best performing system reported so far showing exceptional activity and selectivity even at elevated reaction temperature. Its outstanding performance has been demonstrated on the synthesis of several high value fine chemicals including biopesticides, fragrances and APIs. This work is of outstanding importance, as stereoselective catalyst development opens new directions in the sustainable and economical synthesis of fine chemicals, one of the main topics being, for example, sustainable agriculture. The stability of the Z-selective catalysts – especially for ruthenium systems – is extremely important as upon catalyst decomposition Ru-H species may form initiating double bond isomerization. This undesired feature can significantly decrease the catalyst selectivity resulting in the contamination of the reaction product. This is one of major problem for example in case of biopesticide synthesis when even traces of contaminants can significantly decrease the effectiveness of the pheromones. The reported new catalyst – which was designed by theoretical tools – is able to minimize this issue and provide a gap-filling solution for the synthesis of these high value fine chemicals.

This manuscript is a high-quality document, the scientific work is well-designed and performed at high level. The quality of the work is raised by the fact that, in addition to the preparative work, supporting detailed theoretical calculations were also performed. Although, some typos can be found in the documents it is suggested to be accepted for publication as is (These typos can be fixed during the galley corrections).

Page 3: last sentence lower instead of lover

Scheme 1, top left: compound 2 stereochemistry is E, it would be better to draw as Z

Page 5, second paragraph: ...cheap staring material... should be ... cheap starting material...

Page 9, second paragraph: ...set of testes to verify... should be ...set of tests to verify...

Page 11, last paragraph: ... pheromones are biodegradable, are used in small amounts... should be rewording fore example ... pheromones are biodegradable and used in small amounts...

Scheme 5: The 68% yield for 44 is pretty amaizing as upon the cross-metathesis of conjugated diens such as 43 a mixture of homologues could be expected.

Reviewer #2 (Remarks to the Author):

In this manuscript, Grela, Kajetanowicz, Poater, and co-workers present the preparation and catalytic activity of a ruthenium olefin metathesis catalyst (Ru3) that achieves high selectivity for Z-alkenes under harsh conditions. Building on seminal work by Grubbs and others (Ru1 and Ru2), Z-selective olefin metathesis has attracted significant attention for its ability to form carbon-carbon double bonds with a high preference for the Z (cis) isomer. Catalysts capable of selectively producing Z-olefins are invaluable in various fields, including natural product synthesis, pharmaceuticals, and material science. In contrast to academic laboratories, in an industrial setting, olefin metathesis catalysts must meet a comprehensive set of requirements, including catalyst loading, stability, viscosity, concentration, and reaction temperature.

In line with this reasoning, the authors disclose that an ingenious modification of the dithiolate ligand on an existing Z-catalyst scaffold yields a very stable and efficient catalyst meeting industrial needs. As expected from these authors, the manuscript is well-written and the study is quite thorough. Similarly, the ESI is exemplary. Altogether, this is a very good manuscript, and I commend the authors for presenting such a nice study. However, despite these positive comments, I am a little concerned that the work is still not novel enough to meet the scope/visibility of this journal.

For one, the work involves the modification of a known catalyst (Ru2), which has already been shown to perform well in Z-selective olefin metathesis (albeit under mild conditions). To illustrate this point, consider Yuzu lactone 11, which Grubbs and co-workers obtained (Ref 28) in 68% yield (E/Z: 95/5) after 1 hour at 40°C in CH₂Cl₂ (3 mM) using Ru2 (6 mol%). In comparison, the same lactone is obtained in 66% yield (E/Z: 99/1) after 8 hours at 110°C in PAO6 Polyalphaolefin 6 (0.2 M) using Ru3 (1 mol%). While the elevated temperature and concentration are important improvements for industrial needs, these hardly offset the fact that the same reaction has been achieved faster under mild conditions.

In another example, comparing the reactivity of Mo1 and Ru2 to that of Ru3 at these elevated conditions for the formation of Civetone 13 (Scheme 3), the authors conclude that Ru3 is a much better catalyst at elevated temperature, pointing to a degradation of Mo1 and Ru2. It should be noted that Civetone 13 has also been obtained in 44% yield (E/Z: 95/5) in a continuous flow reactor using a variant of Ru1 (7.5 mol%; 3 mM in DCE) after 3 hours at 70°C (Ref 34; not discussed herewith). While Ru3 does improve the yield and selectivity of the reaction with respect to Ru1 at a lower catalyst loading (0.5 mol%, 76% yield, 98:2 E/Z ratio), one can still question if this improvement is significant in the context of this journal.

Finally, I commend the authors for the extensive work presented in Scheme 4 which showcases the activity of Ru3 under classical conditions. However, Scheme 4 also demonstrates that this new catalyst, both in selectivity and reactivity, is comparable to Ru2 with no significant improvements.

All in all, I would happily support the publication of this manuscript in this journal provided the authors could demonstrate that this new catalyst framework is able to reach reactivities that are reputedly challenging in this field both under academic (mild) and industrial (harsh) conditions. For example, one could consider sterically hindered substrates at the alpha-position to the alkene but of course other options could be considered.

Here are a few minor suggestions for the authors to consider:

1. There are several English/style typos that will require additional proofreading. A few examples are provided below:

- a. "can come at the cost of lover catalyst activity"
- b. "a separate set of testes"
- c. "to decomposition"
- d. Scheme 6: C' the Ph- should be in red.

2. Catalyst stability (Scheme 2. C): Since the manuscript emphasizes the propensity of the catalyst to perform under harsh conditions (i.e., 110°C), it would have been interesting to also evaluate the stability of Ru3 vs. Ru2 at 110°C (note that THF can safely be taken to 110°C in a sealed J-Young NMR tube). Similarly, It would have been interesting to also compare the thermal stability of Ru3 to that of Ru1 and Mo1 both at 60°C and at 110°C.

3. The following sentence is not clear: "The key CNHC-Ru-S1 angle, which is representative of the trans effect and thus predisposition of the catalyst to decomposition through the 1,2-shift amounted to 152.21° and 153.76°."

4. Scheme 3: I invite the authors to provide a different figure for Ru3. At present, this figure suggests that the IPrO- is trans to the carbene, which contradicts the X-ray data.

Reviewer #3 (Remarks to the Author):

In this manuscript, the authors report on the synthesis of a novel, well-defined ruthenium-alkoxybenzylidene initiator for Z-selective olefin metathesis. This new catalyst precursor is based on a known scaffold that relies on the use of a chelating dithiolate ligand. The first representative of this family of compounds was reported by Hoveyda et al. in 2013. Further investigations and improvements came from several other research groups, including those of Grubbs, Pederson et al., and Mauduit et al. The present work discloses a further variation on the dithiolate unit, more specifically, the introduction of a dithioquinoxazoline function. Due to the presence of two nitrogen atoms in its heterocyclic core, this ligand displayed a weaker nucleophilicity than the current state-of-the-art dichlorocatecholthiolate. This feature was anticipated to minimize catalyst decomposition via a 1,2-sulfide shift mechanism, which should allow to perform reactions at higher temperatures.

Although DFT studies carried out to compare the deactivation paths of the novel catalyst precursor reported in this study and its predecessor were not fully conclusive, catalytic tests clearly evidenced that the use of a dithioquinoxazoline chelate resulted in an improved activity and stereoretention at high temperature, a requisite to achieve the macrocyclic ring-closing metathesis of musk lactones under reactive distillation conditions. Moreover, the activity of the novel catalyst also matched the one exhibited by its predecessor in a wide range of cross-metathesis reactions carried out under conventional experimental conditions.

Altogether, this study discloses a highly valuable addition to the portfolio of ruthenium catalysts available for Z-selective olefin metathesis reactions. The dithioquinoxazoline ligand can be easily prepared on a large scale from cheap starting materials, which opens the door to industrial applications and should ease the access to the important class of Z-alkenes found in highly desired organic products, as illustrated by the successful synthesis of numerous pheromones, musks, and active pharmaceutical ingredients using catalyst loadings of 1 mol% or less. Hence, I recommend publication of this work in Nature Communications.

I have found the manuscript well-crafted and clearly written. The novel catalyst has been fully characterized and its molecular structure was determined by X-ray crystallography. Its catalytic potentials have been evaluated using a broad range of unsaturated substrates under diverse experimental conditions. The discussions are sound and supported by a rich bibliography. The supporting information contains enough details for the work to be reproduced. I would, however, suggest a minor revision to address the following issues:

1. The analytic method used to determine the E/Z ratio is not always specified. In section 4 of the experimental part, it is GC. What about the reactions described in sections 5-8? Is it still GC? Or ^1H NMR?

2. There are a few typos in the text, which would benefit from a careful proofreading, e.g., lover instead of lower on page 3, testes instead of tests on page 9.

3. The snake bites its own tail in Ref. 70.

Reviewer #1 (Remarks to the Author):

This work by Grela describes the synthesis of a new, robust and highly Z selective olefin metathesis catalyst. The reported catalyst is the best performing system reported so far showing exceptional activity and selectivity even at elevated reaction temperature. Its outstanding performance has been demonstrated on the synthesis of several high value fine chemicals including biopesticides, fragrances and APIs. This work is of outstanding importance, as stereoselective catalyst development opens new directions in the sustainable and economical synthesis of fine chemicals, one of the main topics being, for example, sustainable agriculture. The stability of the Z-selective catalysts – especially for ruthenium systems – is extremely important as upon catalyst decomposition Ru-H species may form initiating double bond isomerization. This undesired feature can significantly decrease the catalyst selectivity resulting in the contamination of the reaction product. This is one of major problem for example in case of biopesticide synthesis when even traces of contaminants can significantly decrease the effectiveness of the pheromones. The reported new catalyst – which was designed by theoretical tools – is able to minimize this issue and provide a gap-filling solution for the synthesis of these high value fine chemicals. This manuscript is a high-quality document, the scientific work is well-designed and performed at high level. The quality of the work is raised by the fact that, in addition to the preparative work, supporting detailed theoretical calculations were also performed. Although, some typos can be found in the documents it is suggested to be accepted for publication as is (These typos can be fixed during the galey corrections).

KG: Thank you for your opinion and suggested acceptance to Nature Communications “as it is”. **All minor revisions suggested below were applied.** Thank you for pointing them out.

Page 3: last sentence lower instead of lover

Scheme 1, top left: compound 2 stereochemistry is E, it would be better to draw as Z

Page 5, second paragraph: ...cheap staring material... should be ... cheap starting material...

Page 9, second paragraph: ...set of testes to verify... should be ...set of tests to verify...

Page 11, last paragraph: ... pheromones are biodegradable, are used in small amounts... should be rewording fore example ... pheromones are biodegradable and used in small amounts...

KG: All the above typos and mistakes **have been corrected.** Thank you.

Scheme 5: The 68% yield for 44 is pretty amaizing as upon the cross-metathesis of conjugated diens such as 43 a mixture of homologues could be expected.

KG: Thank you for asking this question. **Yes, we are sure that the reaction occurred selectively** on a terminal C–C double bond. The GC (we used a precise FAME-dedicated GC column) does not show the presence of other products (reaction is surprisingly “clean” and less than 0.8% of non-selective metathesis product is observed), and the retention time of the product peak finely makes the retention time of the authentic pheromone sample (made according to the published literature method using Wittig reaction).

In addition, we performed an independent experiment in which oct-7-en-1-yl acetate (terminal alkene) was subjected to cross-metathesis with Z-3-hexene, as the the product of this reaction corresponds to the product of the non-selective metathesis reaction (hypothetical cross metathesis reaction of the internal double bond instead of terminal one). Having standard sample of this “wrong” product, we looked for it in GC trace of our reaction mixture. Again, such product was not present more than traces, thus excluding not-selective CM of 43 with Z-3-hexene in presence of **Ru3**.

Reviewer #2 (Remarks to the Author):

In this manuscript, Grela, Kajetanowicz, Poater, and co-workers present the preparation and catalytic activity of a ruthenium olefin metathesis catalyst (Ru3) that achieves high selectivity for Z-alkenes under harsh conditions. Building on seminal work by Grubbs and others (Ru1 and Ru2), Z-selective olefin metathesis has attracted significant attention for its ability to form carbon-carbon double bonds with a high preference for the Z (cis) isomer. Catalysts capable of selectively producing Z-olefins are invaluable in various fields, including natural product synthesis, pharmaceuticals, and material science. In contrast to academic laboratories, in an industrial setting, olefin metathesis catalysts must meet a comprehensive set of requirements, including catalyst loading, stability, viscosity, concentration, and reaction temperature.

In line with this reasoning, the authors disclose that an ingenious modification of the dithiolate ligand on an existing Z-catalyst scaffold yields a very stable and efficient catalyst meeting industrial needs. As expected from these authors, the manuscript is well-written and the study is quite thorough. Similarly, the ESI is exemplary. Altogether, this is a very good manuscript, and I commend the authors for presenting such a nice study. However, despite these positive comments, I am a little concerned that the work is still not novel enough to meet the scope/visibility of this journal.

KG: Thank you for your valuable opinion and nice words about the general scientific quality of our work. In the following, we explain why we believe that the present system brings new value and opens new possibilities in organic chemistry.

For one, the work involves the modification of a known catalyst (Ru2), which has already been shown to perform well in Z-selective olefin metathesis (albeit under mild conditions). To illustrate this point, consider Yuzu lactone 11, which Grubbs and co-workers obtained (Ref 28) in 68% yield (E/Z: 95/5) after 1 hour at 40°C in CH₂Cl₂ (3 mM) using Ru2 (6 mol%). In comparison, the same lactone is obtained in 66% yield (E/Z: 99/1) after 8 hours at 110°C in PAO6 Polyalphaolefin 6 (0.2 M) using Ru3 (1 mol%). While the elevated temperature and concentration are important improvements for industrial needs, these hardly offset the fact that the same reaction has been achieved faster under mild conditions.

KG: We agree, these key literature results were referred to in our manuscript (page 7). Please note, however, that in addition to a six-times higher amount of expensive ruthenium catalysts (which brings higher costs and problems during purification), making Yuzu lactone by the cited method requires much larger amounts of solvent. For example, to make 10 grams of Yuzu lactone one must use 25 L of CH₂Cl₂, which is by the way a restricted solvent (please see: <https://www.printing.org/content/2024/05/10/epa-bans-mostuses-of-methylene-chloride> and <https://www.fda.gov/downloads/drugs/guidances/ucm073395.pdf>). Therefore, I see the cited results as a very important academic milestone (**I agree with the Referee fully in this point!**), but today rather of historical nature, not relevant to present environmental needs.

In another example, comparing the reactivity of Mo1 and Ru2 to that of Ru3 at these elevated conditions for the formation of Civetone 13 (Scheme 3), the authors conclude that Ru3 is a much better catalyst at elevated temperature, pointing to a degradation of Mo1 and Ru2. It should be noted that Civetone 13 has also been obtained in 44% yield (E/Z: 95/5) in a continuous flow reactor using a variant of Ru1 (7.5 mol%; 3 mM in DCE) after 3 hours at 70°C (Ref 34; not discussed herewith). While Ru3 does improve the yield and selectivity of the reaction with respect to Ru1 at a lower catalyst loading (0.5 mol%, 76% yield, 98:2 E/Z ratio), one can still question if this improvement is significant in the context of this journal.

KG: We agree with the Referee that using continuous flow (CF) may be a very good solution to this problem. **In the revised manuscript we decided to discuss this important paper more.** However, the need of using high dilution (3 mM, which translates to 4.5 L of a chlorinated solvent needed to make 10 grams of Z-Civetone) and high loading (7.5 mol%) is still a very serious limitation.

Obviously, macro-RCM is not that easy transformation! (Fogg, *Chem. Rev.* **2009**, *109*, 3783–3816). In this context we like to stress that our work is the very first example in which unsaturated macrocycles are made under conditions of metathetical reactive distillation in high Z-selectivity.

Apparently, the academics must continue working on this, using different approaches (like Mauduit's CF, our reactive distillation, immobilization, others), and we see our current manuscript as one of such endeavors.

Finally, I commend the authors for the extensive work presented in Scheme 4 which showcases the activity of Ru₃ under classical conditions. However, Scheme 4 also demonstrates that this new catalyst, both in selectivity and reactivity, is comparable to Ru₂ with no significant improvements.

KG: **It is true**, and we are not trying to hide this. In fact the following passage in our Conclusions: *“Importantly, this unique trait did not come at the cost of limited general usability under 'classical' conditions. On the contrary, the new complex was found to match the activity of known stereoretentive catalysts such as Ru₂ even in the case of complex polyfunctional substrates.”* **refers precisely to what Referee rightly wrote about the results presented in Scheme 4.**

In short, we now have a system that is in spades better in one application (and the ligand is easier to prepare, too) while in other applications it matches the known one.

All in all, I would happily support the publication of this manuscript in this journal provided the authors could demonstrate that this new catalyst framework is able to reach reactivities that are reputedly challenging in this field both under academic (mild) and industrial (harsh) conditions. For example, one could consider sterically hindered substrates at the alpha-position to the alkene but of course other options could be considered.

KG: We are happy to do additional experiments to convince Referee about novelty and usefulness of the new system. However, the proposed metathesis of sterically hindered substrates at the alpha-position requires, as we checked in literature the stereoretentive catalysts with very “small” *o*-fluoro substituted NHC ligands, like the ones reported by Grubbs (e.g. *J. Am. Chem. Soc.* **2017**, *139*, 15640), so our catalyst bearing bulky SIPr ligand would not work here. However, we performed one such experiment, using 3,3-dimethylbut-1-ene but have not observed any reactivity. (By the way, we plan to prepare some small-NHC catalysts in the future, mostly for *E*-selective metathesis, but this work has not even started yet.)

Instead, we can propose for the Referee's judgment a stereoselective CM reaction that shows again the difference between known Mo₁, Ru₂ and our new catalysts (Ru₃ and more). It is the reaction shown below and in Confidential Materials for Referees, and it shows—we hope—the virtues of Ru₃ even at lower temperatures. However, as these are the results considered for a patent and are not yet ready to be disclosed, we kindly ask Referee to keep this in confidence.

Here are a few minor suggestions for the authors to consider:

1. There are several English/style typos that will require additional proofreading. A few examples are provided below:

- a. "can come at the cost of lower catalyst activity"
- b. "a separate set of testes"
- c. "to decomposition"
- d. Scheme 6: C' the Ph- should be in red.

KG: Thank you! **All this is now corrected.**

2. Catalyst stability (Scheme 2. C): Since the manuscript emphasizes the propensity of the catalyst to perform under harsh conditions (i.e., 110°C), it would have been interesting to also evaluate the stability of Ru3 vs. Ru2 at 110°C (note that THF can safely be taken to 110°C in a sealed J-Young NMR tube). Similarly, It would have been interesting to also compare the thermal stability of Ru3 to that of Ru1 and Mo1 both at 60°C and at 110°C.

KG: Following the request of Referee, **these additional measurements at 110 °C have been done**, and we found even higher difference between stability of **Ru2** (less stable) and **Ru3** (much more stable). **This chart is added now to Scheme 2**, and included in Confidential Materials for Referees. To be honest we were too timid to do it, and without the Referee suggestion we would never attempt it. *So, we are very grateful to Referee for suggesting us doing this!*

3. The following sentence is not clear: "The key CNHC-Ru-S1 angle, which is representative of the trans effect and thus predisposition of the catalyst to decomposition through the 1,2-shift amounted to 152.21° and 153.76°."

KG: Thank you! **This sentence was rewritten to make it more clean.**

4. Scheme 3: I invite the authors to provide a different figure for Ru3. At present, this figure suggests that the IPrO- is trans to the carbene, which contradicts the X-ray data.

KG: Thank you! **Drawing of Ru3 was altered to make it more close to the XRD structure.**

Reviewer #3 (Remarks to the Author):

In this manuscript, the authors report on the synthesis of a novel, well-defined ruthenium-alkoxybenzylidene initiator for Z-selective olefin metathesis. This new catalyst precursor is based on a known scaffold that relies on the use of a chelating dithiolate ligand. The first representative of this family of compounds was reported by Hoveyda et al. in 2013. Further investigations and improvements came from several other research groups, including those of Grubbs, Pederson et al., and Mauduit et al. The present work discloses a further variation on the dithiolate unit, more specifically, the introduction of a dithioquinoxazoline function. Due to the presence of two nitrogen atoms in its heterocyclic core, this ligand displayed a weaker nucleophilicity than the current state-of-the-art dichlorocatecholthiolate. This feature was anticipated to minimize catalyst decomposition via a 1,2-sulfide shift mechanism, which should allow to perform reactions at higher temperatures.

Although DFT studies carried out to compare the deactivation paths of the novel catalyst precursor reported in this study and its predecessor were not fully conclusive, catalytic tests clearly evidenced that the use of a dithioquinoxazoline chelate resulted in an improved activity and stereoretention at high temperature, a requisite to achieve the macrocyclic ring-closing metathesis of musk lactones under reactive distillation conditions. Moreover, the activity of the novel catalyst also matched the one exhibited by its predecessor in a wide range of cross-metathesis reactions carried out under conventional experimental conditions.

Altogether, this study discloses a highly valuable addition to the portfolio of ruthenium catalysts available for Z-selective olefin metathesis reactions. The dithioquinoxazoline ligand can be easily prepared on a large scale from cheap starting materials, which opens the door to industrial applications and should ease the access to the important class of Z-alkenes found in highly desired organic products, as illustrated by the successful synthesis of numerous pheromones, musks, and active pharmaceutical ingredients using catalyst loadings of 1 mol% or less. Hence, I recommend publication of this work in Nature Communications.

I have found the manuscript well-crafted and clearly written. The novel catalyst has been fully characterized and its molecular structure was determined by X-ray crystallography. Its catalytic potentials have been evaluated using a broad range of unsaturated substrates under diverse experimental conditions. The discussions are sound and supported by a rich bibliography. The supporting information contains enough details for the work to be reproduced. I would, however, suggest a minor revision to address the following issues:

KG: Thank you for your opinion and the suggestion of acceptance to Nature Communications. **All the minor revisions have been applied accordingly.** Thank you for pointing them out.

1. The analytic method used to determine the E/Z ratio is not always specified. In section 4 of the experimental part, it is GC. What about the reactions described in sections 5-8? Is it still GC? Or ¹H NMR?

KG: Good point. **It is now corrected in the revised manuscript and in SI.**

2. There are a few typos in the text, which would benefit from a careful proofreading, e.g., lover instead of lower on page 3, testes instead of tests on page 9.

3. The snake bites its own tail in Ref. 70.

KG: Good point! **It is now corrected in the revised manuscript and in SI.**

* * *

We are very grateful to all three Referees for their truly useful and fair comments!

REVIEWERS' COMMENTS

Reviewer #2 (Remarks to the Author):

I apologize to the authors for the delay in reviewing their corrections due to unforeseen personal circumstances.

I commend the authors for thoroughly addressing all of my initial concerns, and I am pleased to support the publication of this revised manuscript in Nature Communications.

It is an excellent article—congratulations to the team.